# From Spermiogram to Bio-Functional Sperm Parameters: When and Why Request Them?

**DOI:** 10.3390/jcm9020406

**Published:** 2020-02-03

**Authors:** Rosita A. Condorelli, Aldo E. Calogero, Giorgio I. Russo, Sandro La Vignera

**Affiliations:** 1Department of Clinical and Experimental Medicine, University of Catania, 95123 Catania, Italy; rosita.condorelli@unict.it (R.A.C.); acaloger@unict.it (A.E.C.); 2Department of Surgery, Urology Section, University of Catania, 95123 Catania, Italy; giorgioivan1987@gmail.com

**Keywords:** bio-functional sperm parameters, sperm mitochondrial function, sperm parameters, male infertility

## Abstract

The aim of this experimental study was to evaluate whether infertile patients may benefit from the evaluation of bio-functional sperm parameters in addition to the conventional semen analysis. To accomplish this, we evaluated the correlation between conventional and bio-functional sperm parameters based on their percentile distribution in search of a potential threshold of these latter that associates with conventional sperm parameter abnormalities. The study was conducted on 577 unselected patients with infertility lasting at least 12 months. We identified cut-off values according to the median of the population for mitochondrial membrane potential (MMP), number of alive spermatozoa, and chromatin abnormality. High MMP (HMMP) (≥46.25%) was associated with sperm concentration, sperm count, progressive motility, and normal form. Low MMP (LMMP) (≥36.5%) was found to be associated with semen volume, sperm concentration, total sperm count, progressive motility, total motility, and normal form. The number of alive spermatozoa (≥71.7%) was associated with sperm concentration and progressive motility whereas abnormal chromatin compactness (≥21.10%) was associated with sperm concentration, total sperm count, and progressive motility. The data would suggest that, for every increase in the percentile category of sperm concentration, the risk of finding an HMMP≤46.25 is reduced by 0.4 and by 0.66 for a total sperm count. This risk is also reduced by 0.60 for every increase in the percentile category of sperm progressive motility and by 0.71 for total sperm motility. Each increment of percentile category of the following sperm parameter was followed by a decrease in the risk of finding an LMMP≤36.5: sperm concentration 1.66, total sperm count 1.28, sperm progressive motility 1.27, total sperm motility 1.76, and normal form 1.73. Lastly, the data showed that, for every increase in the percentile category of total sperm count, the risk of finding an abnormal chromatin compactness ≤21.10 is reduced by 1.25 (1.04–1.51, *p* < 0.05) and an increase of total sperm motility is associated with a reduced risk by 1.44 (1.12–1.85, *p* < 0.05). Results suggest a correlation between bio-functional and conventional sperm parameters that impact the sperm fertilizing potential. Therefore, the evaluation of bio-functional sperm parameters by flow cytometry may be useful to explain some cases of idiopathic male infertility.

## 1. Introduction

According to the main guidelines, semen analysis represents the first level examination for male infertility in the clinical practice [1,2,3]. However, it has some limitations including the paucity of information on sperm functional capacity. The evaluation of bio-functional sperm parameters is not suggested for all infertile patients, even if they can explain some cases of apparently idiopathic male infertility such as in normozoospermia patients.

Several bio-functional sperm parameters have been evaluated and correlated with conventional sperm parameters such as mitochondrial membrane potential (MMP) and sperm DNA fragmentation, which have, so far, shown the greatest evidence in the literature.

A large number of studies have evaluated the reproductive consequences of an elevated number of spermatozoa with fragmented DNA without looking for a correlation with an alteration of conventional sperm parameters. Moreover, these studies have investigated the relationship between DNA damage and reproductive outcomes both in spontaneous pregnancies and in assistive reproductive techniques. The results suggest that DNA damage is associated with a lower pregnancy rate in natural fertilization, intrauterine insemination (IUI), and intracytoplasmic sperm injection (ICSI) and that patients with a low percentage of spermatozoa with DNA fragmentation have a higher pregnancy rate compared to patients with elevated levels of the same parameter who underwent IUI [4]. The association between DNA damage and reproductive outcomes following ICSI is less clear, even though, in a recently published meta-analysis, a negative impact of DNA fragmentation was shown with the use of this technique [5].

Various studies highlight the correlation between impaired mitochondrial function, as shown by low mitochondrial membrane potential (MMP) and decreased sperm motility [6,7,8]. Subsequently, a positive correlation between MMP, total motility, and sperm vitality was reported in 230 men [9].

Bio-functional sperm parameters could be altered in andrological and systemic diseases, but only in a few specific conditions the evaluation of sperm DNA fragmentation is suggested in the clinical practice. These include varicocele, idiopathic infertility, recurrent miscarriages, ICSI failures cycles, an incorrect lifestyle, and exposure to environmental risk factors [10].

To date, however, it is not known when, in the presence of conventional sperm parameter abnormalities, to proceed with the evaluation of bio-functional sperm parameters in the attempt to understand the true cause of idiopathic male infertility. Therefore, the purpose of this study was to evaluate whether infertile patients may benefit from the evaluation of bio-functional sperm parameters in addition to the conventional semen analysis. To accomplish this, we evaluated the correlations between conventional and bio-functional sperm parameters, based on their percentile distribution, in an attempt to identify potential bio-functional sperm parameter thresholds that associates with conventional sperm parameter abnormalities. This was achieved by evaluating which conventional sperm parameters can intercept a bio-functional sperm alteration to know when to request this latter type of analysis to patients with idiopathic infertility.

## 2. Materials and Methods

### 2.1. Patient Selection

The conventional and bio-functional sperm parameters of 577 unselected patients with infertility lasting longer than 12 months were evaluated. In this group, the female factor determining infertility was excluded (regular ovulation and tubal patency, absence of endometrial diseases, absence of cervico-vaginal infections). The Ethics Committee of University teaching Hospital of “Policlinico-Vittorio Emanuele”, University of Catania, (Catania, Italy) approved this study. All methods were performed in accordance with the more relevant guidelines [11,12] and regulations. All participants were asked for and provided their informed consent.

### 2.2. Experimental Design 

Each patient enrolled in this study underwent semen analysis for the evaluation of conventional and bio-functional sperm parameters by flow cytometry. These latter include MMP, degree of chromatin compactness, sperm apoptosis/vitality, and DNA fragmentation.

### 2.3. Semen Analysis for Conventional Sperm Parameter Evaluation

Semen analysis was conducted according to the WHO criteria [11].

## 3. Evaluation of Bio-Functional Sperm Parameters

Flow cytometry analysis was performed using flow cytometer EPICS XL (Coulter Electronics, IL, Milan, Italy) equipped with an argon laser at 488 nm. We used the FL1 detectors for the green (525 nm), FL2 for the orange (575 nm), and FL3 for the red (620 nm) fluorescence. Furthermore, 100,000 events (low velocity) were measured for each sample and analyzed by the software Sistem II ™, Version 3.0 (SPSS Inc., Chicago, IL, USA).

### 3.1. Evaluation of the Mitochondrial Membrane Potential

The damage of MMP is an early event of the apoptosis and it is reversible. MMP can be evaluated using the lipophilic probe 5,5′,6,6′-tetrachloro-1,1′,3,3′tetraethyl-benzimidazolylcarbocyanine iodide (JC-1). JC-1 is able to penetrate selectively in mitochondria and it exists in monomeric form, which emits at 527 nm. Following excitation at 490 nm and in relation to the membrane potential, JC-1 is able to form aggregates emitting at 590 nm. Therefore, the fluorescence changes reversibly from green to orange as soon as the mitochondrial membrane becomes more polarized. In viable cells with a normal membrane potential, JC-1 is in the mitochondrial membrane in the form of aggregates emitting in an orange fluorescence, while, in the cells with membrane potential, it remains in the cytoplasm in a monomeric form, which gives a green fluorescence.

An aliquot containing 1 × 10^6^/mL of spermatozoa were incubated with JC-1 in the dark, for 10 minutes, at a temperature of 37 °C. At the end of the incubation period, the cells were washed in PBS and analyzed as previously published [13].

### 3.2. Assessment of the Degree of Chromatin Compactness

The evaluation of chromatin integrity was performed after permeabilization of the cell membrane to allow the access of the fluorophore within the nucleus. An aliquot of 1 × 10^6^ spermatozoa was incubated with LPR DNA-Prep Reagent containing 0.1% potassium cyanate, 0.1% NaN_3_, non-ionic detergents, saline, and stabilizers (Beckman Coulter, IL, Milan, Italy) in the dark at room temperature for 10 minutes and then further incubated with Stain DNA-Prep Reagent containing 50 µg/mL of propidium iodide (PI) (<0.5%), RNase A (4 KUnitz/mL), <0.1% NaN3, saline, and stabilizers (Beckman Coulter, IL, Milan, Italy) in the dark at room temperature. Flow cytometry analysis was performed after 30 minutes using the FL3 detector [13].

### 3.3. Evaluation of Sperm Apoptosis/Vitality

The externalization of phosphatidylserine (PS) on the outer cell surface is an early signal of apoptosis. The assessment of PS externalization was performed using annexin V, which is a protein that binds selectively to PS in the presence of calcium ions, and was FITC-labeled. During apoptosis, the cells exhibited the PS even before the loss of semi-permeability. Therefore, marking simultaneously the cells with annexin V and PI, we could distinguish: alive (with intact cytoplasmic membrane), apoptotic, or necrotic cells. Staining with annexin V and PI was obtained using a commercially available kit (Annexin V-FITC Apoptosis, Sigma Chemical (Beckman Coulter, IL, Milan, Italy)).

An aliquot containing 0.5 × 10^6^/mL was suspended in 0.5 mL of buffer containing 10 µL of annexin V-FITC and 20 µL of PI and incubated for 10 minutes in the dark. After incubation, the sample was analyzed immediately by the detectors FL-1 (FITC) and FL3 (PI) [13]. The different pattern of staining allowed us to identify the different cell populations. FITC negative and PI negative indicate viable cells, FITC positive and PI negative indicate cells in early apoptosis with cytoplasmic membrane still intact, and FITC positive and PI positive indicate cells in late apoptosis.

### 3.4. Assessment of DNA Fragmentation

The evaluation of DNA fragmentation was performed by the TUNEL method. This uses the TdT (Terminal deoxynucleotidyl Transferase), which is an enzyme that polymerizes, at the level of DNA breaks and modified nucleotides conjugated to a fluorochrome. The TUNEL assay was performed by using a commercially available kit (Apoptosis Mebstain kit, Beckman Coulter, Milan, Italy). To obtain a negative control, TdT was omitted from the reaction mixture. The positive control was obtained pretreating spermatozoa (about 0.5 × 10^6^) with 1 mg/mL of deoxyribonuclease I, not containing RNAse, at 37 °C for 60 min prior to staining. The reading was performed by flow cytometry using the FL1 detector [13].

## 4. Statistical Analysis

Continuous variables are presented as median and an interquartile range (IQR) and were compared by the Student’s independent t-test or the Mann-Whitney U test based on their normal or not-normal distribution, respectively (normality of variables’ distribution was tested by the Kolmogorov-Smirnov test). Categorical variables were tested with the Chi-square test. Univariable and multivariable logistic regression analyses assessed the association between conventional and non-conventional sperm parameters. Furthermore, conventional parameters were categorized according to the WHO percentiles (1°, 5°, 10°, 25°, 50°, 75°, 90°, and 95° percentile). Univariate logistic regression analysis has been performed to verify associations between non-conventional sperm parameters and conventional percentile groups. Thresholds used for the analysis have been calculated based on the median distribution in our population.

All statistical analyses were completed using Stata software, version 14 (Stata Corp LP. 2015. Stata Statistical Software: Release 14. College Station, TX, USA). For all statistical comparisons, a significance level of *p* < 0.05 was considered to show differences between the groups.

## 5. Results

Table 1 shows the univariate linear regression analysis comparing conventional and bio-functional sperm parameters. We identified a threshold of bio-functional sperm parameters, according to the median of the population only for HMMP, LMMP, alive spermatozoa, and chromatin compactness.

Table 2 shows the association between these cut-offs and conventional sperm parameters at univariate logistic regression analysis. Threshold for sDNAfrag was not given due to lack of association for all comparisons.

In particular, HMMP (≥46.25%) was associated with sperm concentration (Odds ratio (OR): 0.141 (95% CI 0.087–0.227); *p* < 0.01), total sperm count (OR: 0.154; 95% CI 0.079–0.300); *p* < 0.01), sperm progressive motility (OR: 0.246 (95% CI 0.147–0.412), *p* < 0.01) and normal forms (OR: 0.329; 95% CI 0.158–0.638, *p* < 0.01). LMMP (≥36.5%) was found to be associated with semen volume (OR: 2.282; 95% CI 1.048–4.971, *p* < 0.05), sperm concentration (OR: 5.816; 95%CI 3.687–9.176, *p* < 0.01), total sperm count (OR: 3.316; 95% CI 1.872–5.876, *p* < 0.01), sperm progressive motility (OR: 3.944; 95%CI 2.407–6.460, *p* < 0.01), total sperm motility (OR: 1.847; 95% CI 1.034–3.297; *p* < 0.01) and normal form (OR: 3.964 (95%CI 1.832–8.579), p < 0.01).

Alive spermatozoa (≥71.7%) were associated with sperm concentration (OR: 0.360; 95% CI 0.236–0.548), *p* < 0.01) and sperm progressive motility (OR: 0.589; 95%CI 0.374–0.927, *p* < 0.05). 

Abnormal chromatin compactness (≥21.1%) was associated with sperm concentration (OR: 2.938; 95% CI 1.933–4.465, *p* < 0.01), total sperm count (OR: 2.340; 95%CI 1.385–3.954, *p* < 0.01), and sperm progressive motility (OR: 2.333; 95%CI 1.478–3.682, *p* < 0.01). 

Table 3 shows the variations of conventional sperm parameters on the basis of percentiles in ascending order (1°, 5°, 10°, 25°, 50°, 75°, 90°, 95° percentile) according to the cut-offs found. 

The data show that, for every increase in the percentile category of sperm concentration, the risk of finding an HMMP ≤46.25% was reduced by 0.4 (0.22–0.75, *p* < 0.05) and by 0.66 (0.52–0.62, *p* < 0.05) for total sperm count. This risk is also reduced by 0.60 (0.32–1.07, *p* < 0.05) for every increase in the percentile category of sperm progressive motility and by 0.71 (0.54–0.93, *p* < 0.05) for total sperm motility. Each increment of percentile category of the following sperm variables is followed by a reduction in the risk of finding an LMMP ≤36.5%: sperm concentration 1.66 (1.12–2.45) *p* < 0.05, total sperm count 1.28 (1.06–1.53) *p* < 0.05), sperm progressive motility 1.27 (0.99–1.62) *p* < 0.05), total sperm motility 1.76 (1.26–2.44) *p* < 0.05), and normal form 1.73 (1.10–2.73) *p* < 0.05. 

Lastly, the data show that, for every increase in the percentile category of total sperm count, the risk of finding an abnormal chromatin compactness ≤21.1% decreases by 1.25 (1.04–1.51, *p* < 0.05) and an increase of total sperm motility is associated with a reduced risk of 1.44 (1.12–1.85, *p* < 0.05) (Figure 1 and Figure 2).

## 6. Discussion

Semen analysis remains the gold standard for the diagnosis of male infertility, which provides useful information on conventional sperm parameters. In this study, we found that bio-functional sperm parameters, which could orient the clinician toward a diagnosis of the specific cause, are useful in some cases of idiopathic infertility. We showed a correlation between conventional and bio-functional sperm parameters providing thresholds to know when to request the valuation of bio-functional parameters. 

The correlation between sperm concentration, motility, normal forms, and MMP found in our study suggested that this bio-functional sperm parameter should be required in patients with oligo-astheno-teratozoospermia.

Previous evidence has shown that MMP correlates with the total sperm motility and with sperm apoptosis [9]. The present study not only showed that LMMP correlated with sperm progressive, total motility, and volume, but also that it is possible to identify a threshold of 36.5% above which the probability of finding conventional sperm parameter abnormality increases. We also identified a threshold of HMMP of ≥46.25%.

The reduction of sperm motility due to the decrease of the percentage of alive spermatozoa is a physiological event. Our study has shown that, when the percentage of alive spermatozoa falls below 71.7%, sperm motility and sperm concentration are reduced. It has been reported that caspase enzymatic activity is higher in semen samples with low motility [14]. Moreover, the correlation of PS externalization on the outer cell surface and the loss of sperm MMP seems to be suggestive of an early apoptosis phenotype in sperm subpopulations with low sperm motility [15].

The degree of chromatin compactness damage has been correlated with abnormal sperm concentration, total counts, and progressive motility. We indicated, as a threshold of chromatin compactness, a value of 21.1%, which is above the probability of finding conventional sperm parameter anomalies (i.e., oligozoospermia or asthenozoospermia) increases. Scientific evidence confirms that low sperm motility seems to be associated with sperm chromatin defects such as low sperm genomic integrity and abnormal DNA condensation and by defects of sperm midpieces altering a spermatogenic remodeling process [16].

Lastly, we did not find any correlations between sperm DNA fragmentation and conventional sperm parameters. These data are confirmed by studies showing that the sperm DNA fragmentation test provides clinically relevant information for natural or assisted reproductive techniques (ART) independently of those derived from conventional semen parameters [17]. Other evidence adds that sperm DNA fragmentation may play a role in unexplained recurrent pregnancy loss despite normal conventional sperm parameters [18]. Based on these premises, sperm DNA fragmentation has been recommended as diagnostic tests in cases of repeated implantation failure, varicocele surgery before undergoing ART treatment, and/or in patients having unsuccessful ART outcomes despite normal sperm parameters, which may be prior to the decision of testicular sperm extraction for ICSI [19]. On this account, the evaluation of sperm DNA fragmentation should be a more routinely asked test.

On the contrary, we found that some alterations of conventional sperm parameters may lead to the request of more complete diagnostic examinations to trace the causes of male infertility. Higher values of sperm concentration and motility reduces the risk of finding mitochondrial damage and abnormal chromatin compactness. Therefore, in cases of oligozoospermia and asthenozoospermia, a specific therapy aimed at increasing the number of spermatozoa (use of follicle-stimulating hormone, antioxidants, etc.) [20,21] or sperm motility as a prokinetic therapy (myo-inositol, etc.) [22] could improve cases previously considered idiopathic.

According to our data, the percentage of spermatozoa with a normal form may point towards a second level diagnostic testing, such as the evaluation of degree of chromatin compactness, whose causes are often due to an increase in oxidative stress and, therefore, suggest treatment with antioxidants [23].

In conclusion, patients with oligozoospermia or asthenozoospermia may benefit from the evaluations of MMP, while a case of oligozoospermia, asthenozoospermia, or teratozoospermia could support an alteration of sperm chromatin compactness and viability. The identification of the type of abnormality may help in a more suitable and specific therapy prescription. The novelty of this study is a new approach to sperm analysis, which, understandably, must be supported by a standardization of the methods used to evaluate sperm bio-functional parameters.

## Figures and Tables

**Figure 1 jcm-09-00406-f001:**
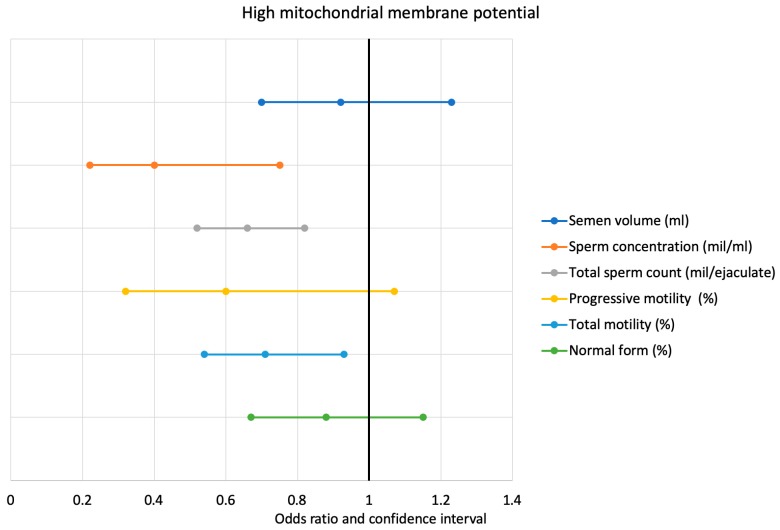
Logistic regression analysis between high mitochondrial membrane potential and percentiles of WHO sperm parameters.

**Figure 2 jcm-09-00406-f002:**
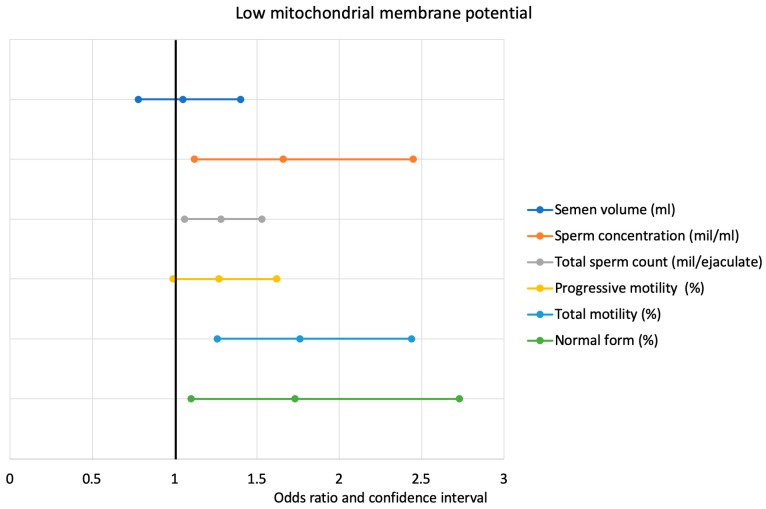
Logistic regression analysis between low mitochondrial membrane potential and percentiles of WHO sperm parameters.

**Table 1 jcm-09-00406-t001:** Univariate linear regression analysis comparing conventional and bio-functional sperm parameters expressed as odds ratio (OR) and a 95% confidence interval.

Bio-Functional Sperm Parameters	Semen volume	Sperm Concentration	Total Sperm Count	Progressive Motility	Total Motility	Normal Form
**HMMP**	0.988 (0.975–1.002)	0.959 (0.95–0.969) **	0.966 (0.954–0.978) **	0.967 (0.958–0.977) **	0.986 (0.975–0.997) *	0.985 (0.972–0.998) *
**LMMP**	1.014 (1.001–1.029) *	1.039 (1.029–1.049) **	1.030 (1.019–1.041) **	1.034 (1.023–1.044) **	1.015 (1.004–1.027) **	1.029 (1.015–1.043) **
**Alive**	1.009 (0.988–1.031)	0.972 (0.96–0.985) **	1.002 (0.988–1.016)	0.977 (0.963–0.991) **	0.984 (0.971–0.998) *	0.999 (0.982–1.016)
**Early apoptosis**	1.004 (0.956–1.055)	1.043 (1.008–1.079) *	1.015 (0.982–1.049)	1.034 (0.994–1.075)	0.994 (0.956–1.034)	0.949 (0.885–1.017)
**Late apoptosis**	0.999 (0.965–1.034)	0.991 (0.971–1.010)	0.993 (0.963–1.019)	1.024 (0.999–1.050)	1.015 (0.992–1.039)	0.992 (0.961–1.025)
**Necrosis**	0.986 (0.960–1.013)	1.037 (1.021–1.054) **	0.998 (0.982–1.015)	1.014 (0.998–1.031)	1.018 (1.002–1.035) *	1.014 (0.995–1.034)
**Chromatin compactness**	1.013 (0.984–1.041)	1.049 (1.028–1.070) **	1.048 (1.027–1.069) **	1.056 (1.031–1.082) **	1.024 (1.002–1.045) *	1.029 (1.005–1.055) *
**DNA fragmentation**	0.964 (0.901–1.031)	0.987 (0.964–1.012)	1.002 (0.976–1.028)	1.023 (0.989–1.058)	0.989 (0.956–1.024)	0.988 (0.948–1.031)

* *p* < 0.05. ** *p* < 0.01.

**Table 2 jcm-09-00406-t002:** Associations between previous cut-offs and conventional sperm parameters at univariate logistic regression analysis expressed as odds ratio (OR) and a 95% confidence interval (CI).

Bio-Functional Sperm Parameters	Semen Volume	Sperm Concentration	Total Sperm Count	Progressive Motility	Total Motility (%)	Normal Form (%)
**HMMP (≥46.25%), yes vs. no**	0.521 (0.246–1.102)	0.141 (0.087–0.227) **	0.154 (0.079–0.300) **	0.246 (0.147–0.412) **	0.600 (0.335–1.074)	0.329 (0.158–0.638) **
**LMMP** **(≥36.5%), yes vs. no**	2.282 (1.048–4.971) *	5.816 (3.687–9.176) **	3.316 (1.872–5.876) **	3.944 (2.407–6.460) **	1.847 (1.034–3.297) *	3.964 (1.832–8.579) **
**Alive** **(≥71.7%), yes vs. no**	1.826 (0.880–3.791)	0.360 (0.236–0.548) **	0.975 (0.593–1.601)	0.589 (0.374–0.927) *	0.665 (0.384–1.154)	0.879 (0.464–1.667)
**Chromatin Compactness (≥21.1%), yes vs. no**	1.876 (0.895–3.931)	2.938 (1.933–4.465) **	2.340 (1.385-3.954) **	2.333 (1.478–3.682) **	1.266 (0.731–2.195)	1.644 (0.860–3.143)

* *p* < 0.05. ** *p* < 0.01.

**Table 3 jcm-09-00406-t003:** Variations of conventional sperm parameters on the basis of percentiles in ascending order, according to the cut-offs found.

Bio-Functional Sperm Parameters	Semen Volume (mL)	Sperm Concentration (mil/mL)	Total Sperm Count (mil/ejaculate)	Progressive Motility (%)	Total Motility (%)	Normal Form (%)
**HMMP (≥46.25%)**	0.92 (0.70–1.23)	0.40 (0.22–0.75) *	0.66 (0.52–0.82) *	0.60 (0.32–1.07) *	0.71 (0.54–0.93) *	0.88 (0.67–1.15)
**LMMP** **(≥36.5%)**	1.05 (0.78–1.40)	1.66 (1.12–2.45) *	1.28 (1.06–1.53) *	1.27 (0.99–1.62) *	1.76 (1.26–2.44) *	1.73 (1.10–2.73) *
**Alive (≥71.7%)**	1.07 (0.81–1.42)	0.96 (0.68–1.37) *	1.06 (0.90–1.24)	0.75 (0.48–1.15)	0.86 (0.69–1.06)	0.89 (0.69–1.15)
**Chromatin Compactness (≥21.1%)**	1.05 (0.80–1.38)	1.40 (0.98-1.98)	1.25 (1.04–1.51) *	1.07 (0.82–1.40)	1.44 (1.12–1.85) *	1.03 (0.78–1.35)

* *p* < 0.05.

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
