# Peer review of "From Spermiogram to Bio-Functional Sperm Parameters: When and Why Request Them?"

_jcm, 2020, doi:10.3390/jcm9020406_

Round 1

Reviewer 1 Report

The authors analyze the relation between conventional semen analysis measures and bio-functional analyses of sperm in a large dataset in an attempt to determine when the latter may be useful in the male infertility evaluation.  Expansion of description of the analytic approach and a more clear and through presentation of the results would greatly improve this interesting manuscript.

Methods, Patient selection, line 75: Do the authors mean to communicate here that the 577 subjects were infertile men (“577 unselected patients with infertility”) or that they were male partners in 577 couples with infertility.  The former implies that these were all infertile men, the latter that this group was likely a mix of men with varying fertility, so the difference is important. Methods, Statistical analysis, line 135. The authors pose an interesting question and have amassed substantial data for its exploration.  However, it is difficult to understand their approach to obtaining useful answers from a reading of this section.  As I understand it, they aim to determine the “…bio-functional sperm parameter thresholds that associates with conventional sperm parameter abnormalities.” (line 69, Introduction)  I believe by this that the authors’ intent is to determine how well, and at what level of abnormality the different conventional semen parameters predict bio-functional abnormalities.  The standard way to accomplish this would be to examine receiver-operator characteristics for each of the conventional parameters in relation to accepted thresholds for abnormal for the bio-functional parameters the authors are interested in.  This would not only identify the cutpoint of interest but also the accuracy of any given conventional test in predicting the likelihood of an abnormal results in the bio-functional measure against which it is being tested.  This would seem a useful approach for the authors. Results: Unfortunately, this section is very unclear to me.  For example, I do  not understand what the numbers in Table 1 are.  They relate to regression analysis apparently, so one thought would be that they are r-values, but values greater than 1 exclude this possibility.  It is not clear to me how this table shows how thresholds for bio-functional sperm parameters were ascertained, and moreover, what these values were. Results, as in prior point, Table 2: This table needs to be better explained and the nature of the numbers in it better described. Results, as in point 3, Table 3: Again, I am unable to interpret this table.  I have tried, but I find it inscrutable.  Where are the values for each of the eight enumerated percentiles for each of these parameters.  Is it possible that this data could be better presented graphically? Early in this section, first or second paragraph, it would be helpful if the authors would summarize succinctly the principal findings of the study, with reference to any precedents in the literature.

Author Response

Attached the rebuttal letter

1.Question : Methods, Patient selection, line 75: Do the authors mean to communicate here that the 577 subjects were infertile men (“577 unselected patients with infertility”) or that they were male partners in 577 couples with infertility. The former implies that these were all infertile men, the latter that this group was likely a mix of men with varying fertility, so the difference is important.

Reply: Thanks for your comment. This group is composed by 577 patients with infertility longer than 12 months where the female factor determining infertility was excluded.
We have clarified this aspect in the manuscript (Line 78-80)

2.Question : Methods, Statistical analysis, line 135. The authors pose an interesting question and have amassed substantial data for its exploration. However, it is difficult to understand their approach to obtaining useful answers from a reading of this section. As I understand it, they aim to determine the “…bio-functional sperm parameter thresholds that associates with conventional sperm parameter abnormalities.” (line 69, Introduction) I believe by this that the authors’ intent is to determine how well, and at what level of abnormality the different conventional semen parameters predict bio-functional abnormalities. The standard way to accomplish this would be to examine receiver-operator characteristics for each of the conventional parameters in relation to accepted thresholds for abnormal for the bio-functional parameters the authors are interested in. This would not only identify the cutpoint of interest but also the accuracy of any given conventional test in predicting the likelihood of an abnormal results in the bio-functional measure against which it is being tested. This would seem a useful approach for the authors.

Reply: Thanks for the valuable comment. After statistical revision of the manuscript we confirm the analytical approach provided.

3. Question : Results: Unfortunately, this section is very unclear to me. For example, I do not understand what the numbers in Table 1 are. They relate to regression analysis apparently, so one thought would be that they are r-values, but values greater than 1 exclude this possibility. It is not clear to me how this table shows how thresholds for bio-functional sperm parameters were ascertained, and moreover, what these values were.

Reply: We apologize for the misunderstanding created. However, Table 1 showed the Univariate linear regression analysis comparing conventional and bio-functional sperm parameters expressed as odds ratio (OR) and 95% confidence interval. We corrected the table.

4.Question : Results, as in prior point, Table 2: This table needs to be better explained and the nature of the numbers in it better described.

Reply: Table 2 shows the association between previous cut-offs and conventional sperm parameters at univariate logistic regression analysis expressed as odds ratio (OR) and 95% Confidence interval (CI).

5.Question : Results, as in point 3.

Reply: In this table, conventional parameters were categorized according to the WHO percentiles (1°, 5°, 10°, 25°, 50°, 75°, 90°, 95° percentile). Univariate logistic regression analysis has been performed to verify associations between non-conventional sperm parameters and conventional percentiles groups.

6.Question : Table 3: Again, I am unable to interpret this table. I have tried, but I find it inscrutable. Where are the values for each of the eight enumerated percentiles for each of these parameters. Is it possible that this data could be better presented graphically? Early in this section, first or second paragraph, it would be helpful if the authors would summarize succinctly the principal findings of the study, with reference to any precedents in the literature.

Reply: As previous.
For example, The data showed that for every increase in the percentile category of sperm concentration, the risk of finding a HMMP ≤46.25% reduced by 0.4 (0.22-0.75; p<0.05) and by 0.66 (0.52-0.62; p<0.05) for total sperm count.

Reviewer 2 Report

Reviewers comments:

-the acronym MMP in Abstract needs to be first explained.

-How changes in mitochondrial membrane potential (MMP) can influence sperm production (count, concentration) or morphology? I understand the impact on motility. The other need to be explained.

-Line 75: …unselected male patients….

-Line 75: AA should give the characteristics of these men.

-Line 75: AA should give the cause of infertility of these men.

-Lines 91, 104, 113, 127-Please give Refs for the 4 tests.

-Line 114: I did not find the explanation for the acronym PS (phosphatidylserine).

-Line 220: I did not find the explanation for the acronym FSH (follicle stimulating hormone).

-Line 299: I did not find the explanation for the acronym ART (assisted reproduction technology).

-Cytometry was performed independent for all tests?

-Why to use Cytometry instead of Aniline blue for sperm chromatin maturity?

(https://doi.org/10.1186/s12610-018-0082-2)

-Why to use Cytometry instead of immunofluorescence for apoptosis?

(https://doi.org/10.1093/humrep/deh727; https://doi.org/10.1016/j.rbmo.2009.10.002)

-Why to use Cytometry instead of immunofluorescence for sperm DNA fragmentation?

(https://doi.org/10.1007/s10815-014-0370-5; http://dx.doi.org/10.1016/j.rbmo.2015.06.019)

-Statistical analysis. If AA determined thresholds and discuss about probabilities, why are not presented ROC curves, areas under the curve and the predictive values? Do AA achieved thresholds by other methods? Please explain. Further, in the text AA use terms such as “correlations” but no data and “r” values (and the associated CI and P) are presented.

-Why AA gave thresholds for tests except for sDNAfrag?

-If AA observed absence of relationship to sDNAfrag regarding other AA. This should be explained why.

Author Response

Attached the rebuttal letter

1. Question : the acronym MMP in Abstract needs to be first explained (done line 14-15 of the revised manuscript)

2. Question : How changes in mitochondrial membrane potential (MMP) can influence sperm production (count, concentration) or morphology? I understand the impact on motility. The other need to be explained.

Reply: These aspects can be correlated with the subsequent loss of motility. Various studies highlight the correlation between MMP and decreased total motility and also sperm vitality (Paoli D. et al, 2011 Fertil Steril. 95:2315-2319) (we modified line 59 in the revised manuscript as suggested). The correlation of PS externalization on the outer cell surface and the loss of sperm MMP seems to be suggestive of an early apoptosis phenotype in sperm subpopulations with low sperm motility. On the other hand, the apoptosis process can therefore determine consequent changes in sperm count and morphology. In addition, sperm chromatin defects such as low sperm genomic integrity and abnormal DNA condensation and by defects of sperm midpiece altering spermatogenic remodelling process (Lines 222-225). We suggest in this manuscript that these phenomena can be connected on the basis of the pathophysiological processes described.

3. Question : Line 75: …unselected male patients….

Reply: This group is composed by 577 patients with infertility longer than 12 months where the female factor determining infertility was excluded. We have clarified this aspect in the manuscript (Line 78-80)

4. Question : Line 75: AA should give the characteristics of these men.

Reply: See the reply in the next comment

5. Question : Line 75: AA should give the cause of infertility of these men.

Reply: In this manuscript our aim was to evaluate the correlation between conventional and bio-functional sperm parameters and not to establish how these parameters vary according to a specific pathology (several diseases can alter these parameters as previously described by our group …. Link pubmed)

6. Question : Lines 91, 104, 113, 127-Please give Refs for the 4 tests.

Reply: Done see line…, line…, line…, line.. of the revised manuscript)

7. Question : Line 114: I did not find the explanation for the acronym PS (phosphatidylserine).

Reply: Done, see line 119 of the revised manuscript

8. Question : Line 220: I did not find the explanation for the acronym FSH (follicle stimulating hormone).

Reply: Done, see lines 239-240 of the revised manuscript

9. Question : Line 299: I did not find the explanation for the acronym ART (assisted reproduction technology).

Reply: Done, see lines 227 of the revised manuscript

10. Question : Cytometry was performed independent for all tests?

Reply: Reviewer's comment is appropriate. Ogni parametro è stato analizzato su differenti aliquote cellulari dello stesso campione.

11. Question : Why to use Cytometry instead of Aniline blue for sperm chromatin maturity?
(https://doi.org/10.1186/s12610-018-0082-2)

Reply: Your comment is interesting and still represents a reason for debate among researchers. In truth, there are no international guidelines regarding this aspect demonstrating higher diagnostic specificity among these. All the methods are considered specifics. The most important aspect concerns the reliability of the method in the same laboratory, as in our case, our extensive experience in flow cytometry on seminal parameters. Flow cytometry allows the researcher to measure a greater number of cellular events in the same unit of time.

12. Question : Why to use Cytometry instead of immunofluorescence for apoptosis?
(https://doi.org/10.1093/humrep/deh727; https://doi.org/10.1016/j.rbmo.2009.10.002)

Reply: Please, see above reply. Flow cytometry allows the researcher to measure a greater number of cellular events in the same unit of time.

13. Question : Why to use Cytometry instead of immunofluorescence for sperm DNA fragmentation?
(https://doi.org/10.1007/s10815-014-0370-5; http://dx.doi.org/10.1016/j.rbmo.2015.06.019)

Reply: For the study of DNA fragmentation, the SDF assay confers a relatively high diagnostic accuracy for infertility detection, among which the TUNEL based methodology seems to achieve higher accuracy than the SCD and Comet assays. Moreover, as previously described, TUNEL test by flow cytometry allows the researcher to measure a greater number of cellular events in the same unit of time.

14. Question : Statistical analysis. If AA determined thresholds and discuss about probabilities, why are not presented ROC curves, areas under the curve and the predictive values? Do AA achieved thresholds by other methods? Please explain. Further, in the text AA use terms such as “correlations” but no data and “r” values (and the associated CI and P) are presented.

Reply: We agree with your observation. At first we though to perform such analysis, however, we drive to the conclusion that demonstrating the accuracy of a “non-conventional parameters” against a “conventional” does not lead to the demonstration of infertility. Such findings in fact could have been misleading and so we omitted. Furthermore, our postulation was to verify an association between both group parameters in order to demonstrated that also a different part of the semen can be altered in such cohort of patients.
Table 1 and 2 show the associations using univariate logistic regression analysis expressed as odds ratio (OR) and 95% Confidence interval (CI).

15. Question : Why AA gave thresholds for tests except for sDNAfrag?

Reply: Threshold for sDNAfrag was not given due to lack of association for all comparisons. We updated results section.

16. Question : If AA observed absence of relationship to sDNAfrag regarding other AA. This should be explained why.

Reply: Threshold for sDNAfrag was not given due to lack of association for all comparisons. We updated results section. Unfortunately, we cannot say why the lack of association, may be this parameter does not reflect other convention parameters.

Round 2

Reviewer 1 Report

This report details the relationships between biofunctional and conventional semen parameters in a population of men in infertile relationships in which obvious female factor infertility factors were not identified.  There are indeed relationships that are statistically significant for selected pairs of these measures, but identification of values of conventional measures that might warrant adding biofunctional measures to an assessment that might enhance the evaluation of a male is obscure.

Statistical Analysis, and Tables 1 and 2: Identification of “a threshold of bio-functional sperm parameters according to the median of the population”  for sperm parameters is  a key outcome of this paper yet the process for this determination is not described    How were these thresholds determined? For example, the process for determining the HMMP cutoff  (46.25%) doesn’t seem to be fully  explained.  Does “according to the median” mean “the median”. If so, do the authors consider median values the threshold for abnormal?  Might the authors wish to use generally accepted clinical thresholds instead, or as well?  It would help if the presentation of methods and results addressed this question. Statistical Analysis, and Tables 1 and 2: One might think that the authors wished to determine which levels of conventional parameters predict the likelihood for the various biofunctional sperm parameters to be“abnormal” (e.g. 5th or 95th percentile, or similar), and show how often the various biofunctional sperm measures were correspondingly abnormal in each instance. Results, line 179, and following, and Table 3: The values in this table are not clearly interpretable.  First the percentile values for showing these relationships are not linear. Further, I presume that, for example, the risk of finding and HMMP below cutoff decreases by 40% for each step in the percentile selections the authors have chosen, but it would be more helpful to the reader if the actual risks for abnormal biofunctional measures in relation to  various values of the conventional ones were portrayed, perhaps graphically, for at least some of the relationships that were statistically significant, and that  the authors feel important. Conclusion, and point 3, above. It would be of interest to know if the authors showed whether there are instances when conventional measures are normal, but abnormal biofunctional outcomes might be expected with any particular likelihood.  If only “patients with oligospermia or asthenospermia may benefit from the evaluations of MMP” (etc.), is it not the case that the finding of abnormal biofunctional parameters in these instances is redundant to what is already known about that individual’s fertility?

Author Response

Attached file with asnwers

1. Statistical Analysis, and Tables 1 and 2: Identification of “a threshold of bio-functional sperm parameters according to the median of the population” for sperm parameters is  a key outcome of this paper yet the process for this determination is not described    How were these thresholds determined? For example, the process for determining the HMMP cutoff  (46.25%) doesn’t seem to be fully    Does “according to the median” mean “the median”. If so, do the authors consider median values the threshold for abnormal?  Might the authors wish to use generally accepted clinical thresholds instead, or as well?  It would help if the presentation of methods and results addressed this question.

2. Statistical Analysis, and Tables 1 and 2: One might think that the authors wished to determine which levels of conventional parameters predict the likelihood for the various biofunctional sperm parameters to be“abnormal” (e.g. 5th or 95th percentile, or similar), and show how often the various biofunctional sperm measures were correspondingly abnormal in each instance.

3. Results, line 179, and following, and Table 3: The values in this table are not clearly interpretable. First the percentile values for showing these relationships are not linear. Further, I presume that, for example, the risk of finding and HMMP below cutoff decreases by 40% for each step in the percentile selections the authors have chosen, but it would be more helpful to the reader if the actual risks for abnormal biofunctional measures in relation to  various values of the conventional ones were portrayed, perhaps graphically, for at least some of the relationships that were statistically significant, and that  the authors feel important.

4. Conclusion, and point 3, above. It would be of interest to know if the authors showed whether there are instances when conventional measures are normal, but abnormal biofunctional outcomes might be expected with any particular likelihood. If only “patients with oligospermia or asthenospermia may benefit from the evaluations of MMP” (etc.), is it not the case that the finding of abnormal biofunctional parameters in these instances is redundant to what is already known about that individual’s fertility?

REPLY

1. Yes, we considered, and we applied for threshold the median of our population. Unfortunately, in literature there are not generally thresholds since these parameters are still not used in clinical practice worldwide. We have updated the methods.

2. In these tables, conventional parameters were categorized according to the WHO percentiles (1°, 5°, 10°, 25°, 50°, 75°, 90°, 95° percentile). Univariate logistic regression analysis has been performed to verify associations between non-conventional sperm parameters and conventional percentiles groups.

3. In table 3, dependent variables (1st row) are expressed as categorical according to the median, while covariates variables, intended as conventional parameters, are categorized according to the percentiles (1°, 5°, 10°, 25°, 50°, 75°, 90°, 95°). We have updated the figures.

4. We have updated figures integrated with table 3 in order to better understand results.

Round 3

Reviewer 1 Report

The authors have made refining adjustments to this interesting manuscript.  Despite my persisting concerns, the authors’ data, and its presentation are interesting and add to the literature on evaluation of male fertility

I am afraid I may have been insufficiently clear. To restate my prior, major concern, the authors state the aim was to “evaluate whether infertile patient may benefit from the evaluation of bio-functional sperm parameters in addition to the conventional semen analysis”.  What I understand them to have done, however, is demonstrate correlations between the various parameters considered, and show the likelihood of abnormalities in conventional semen parameters in relation to various degrees of abnormality in bio-functional ones.  One might have thought, given their stated aims, they might have done the opposite.  Because it seems to me the question is:  how often are normal values of conventional measures associated with abnormal bio-functional ones?   And further, at what thresholds of conventional measures might one expect to glean additional information (i.e. find abnormal bio-functional measures) by performing bio-functional tests. I remain unclear as to why median values are accepted as “thresholds”.